# Evaluation of Factors Affecting the Transition to a Circular Economy (CE) in Vietnam by Structural Equation Modeling (SEM)

**Thảo Việt Trần [1], Thảo Hương Phan [1,\*], Anh Thị Trâm Lê [1] and Trang Mai Trần [2]**

[1] Department of Research Administration, Thuongmai University, Ha Noi 10000, Vietnam; tranvietthao@tmu.edu.vn (T.V.T.); letramanh@tmu.edu.vn (A.T.T.L.)
[2] Vietnam Institute of Economics, Vietnam Academy of Social Sciences, Ha Noi 10000, Vietnam; tranmaitrang@iames.gov.vn
\* Correspondence: thaophandhtm@tmu.edu.vn

**Abstract:** Currently, the transition to a circular economy is becoming a development trend of many countries around the world to cope with climate change and reduce carbon emissions. Vietnam is also one of the countries in the process of taking steps to transition to a circular economy. However, to make a successful transition to a circular economy, citizen participation is essential. Thus, the question is, are people ready to participate in the circular economy? Therefore, this study surveyed 431 people regarding their willingness to participate in the circular economy based on the theory of planned behavior and the structural equation model. The results of empirical research have shown that the factors attention to the environment, and attitude towards intention are the factors that have a strong impact on willingness to participate in the circular economy. Based on the given influencing factors, the authors make some policy suggestions for the Vietnamese government in the transition to a circular economy.

**Keywords:** structural equation modeling; Vietnamese people; circular economy; green development; theory of planned behavior

## 1. Introduction

Environmental pollution is getting worse and worse in almost every country globally [1]. Climate change, environmental pollution, resource scarcity, biodiversity loss, and population growth are urgent factors that threaten the earth's life. When facing the increasing risk of climate change, the sustainable development model becomes more and more critical. Many countries worldwide have transitioned to a sustainable development model to balance economic growth, environmental protection, and social welfare. The circular economy (CE) is considered one of the best solutions to support sustainable development [2]. The CE is utilized because the earth is a closed economic system, and all business activities of production and consumption must be based on a secure dual system. In a CE, the economy and the environment are linked together in a closed circle. However, the concept of CE is still controversial. The most widely accepted definition is the flow of resources, and the efficient use of raw materials and energy are core features of this concept. The CE model brings many benefits compared with the traditional linear economic model [3]. For developing countries, the transition to a CE represents the nation's responsibility to address the challenges posed by environmental pollution and climate change while enhancing its capacity and competitiveness of the economy. The CE uses used materials instead of generating waste disposal costs; minimizes the exploitation of natural resources, making the most of the value of resources; and reduces waste and emissions into the environment [4]. For society, the CE helps reduce social costs in managing and protecting the environment and responding to climate change, creating new markets, new

job opportunities, and improving people's health [3]. For businesses, the CE reduces the risks of overproduction and resource scarcity crises, creates motivation to invest, innovates technology, reduces production costs, and increases the supply chain.

After 35 years of renovation, Vietnam has become a bright growth spot in the region and the world with many remarkable achievements [5]. Vietnam's economy has grown in size, but the quality of growth has also improved and, with this, the material and spiritual life has also improved. However, Vietnam faces many challenges in resource depletion, pollution, environmental degradation, and climate change. Currently, Vietnam ranks 4th in the world in plastic waste, with 1.83 million tons/year. The volume of daily-life solid waste generated is more than 61,000 tons/day, of which up to 71% of the total waste volume (equivalent to 43 thousand tons/day) is treated by the burial method [6]. Many resources are currently severely depleted, particularly coal. Since 2015, Vietnam has had to import coal. It is forecasted that by 2030, it will be necessary to import up to 100 million tons of coal per year [7]. According to the World Bank (WB) calculations, environmental pollution could cost Vietnam up to 3.5% of GDP by 2035. In particular, Vietnam is among the countries which are most vulnerable to climate change [8]. It is forecasted that climate change and natural disasters could cause damage to up to 11% of Vietnam's GDP by 2030 [9]. Therefore, to implement sustainable development goals and international commitments, the approach to the model transformation from a "linear economy" to a "circular economy" should be considered a priority in the new development stage of Vietnam. In facing the opportunities brought by the CE, Vietnam has made many positive actions to facilitate the development of the CE model. Over the years, the Vietnamese government has issued many policies to transform the growth model towards sustainability, strengthen the management of natural resources, protect the environment in response to climate change, and increase recycling and reuse of waste [10].

However, the understanding and the willingness of Vietnamese people to participate in the CE is the issue that needs the most attention. Vietnam's transition to a CE cannot succeed without people's willingness to participate. Therefore, this study investigates people's willingness to participate in the CE based on behavioral theory. This study has empirical significance because it uses the theory of planned behavior to quantify the factors that influence the degree of participation in the CE [11]. In addition, this study also contributes theoretically because it provides implications for the government in finding ways to approach people's behavior and attitudes in participating in the CE. Previous studies have focused on the CE, not on the point of view of participants in the CE in Vietnam. Moreover, this study will also provide policy suggestions towards clean and sustainable production in Vietnam. The industrial output of the CE model can help improve productivity and ecological efficiency, enhance environmental management, and facilitate movement towards sustainable development [12]. The transition to a CE requires the participation of almost all segments of society, from consumers to businesses and other stakeholders [2]. Investigating the factors affecting the willingness to participate in the CE will be the motivating factors for the state and the enterprises in making the transition to the CE soon.

## 2. Literature Review

Geng and Doberstein [13] described the CE as a closed circular flow of matter throughout the economic system. This economic development model is approached based on sustainable economic development, protection of the natural environment, and improvement of social welfare. Geng, Zhu, Doberstein, & Fujita [14] argued that CE is a concept related to the 3R principle in the production and use of products. The 3Rs are "reduce," "reuse," and "recycle." Reduction is the reduction of resource consumption and waste production; reuse and recycling involve processing wastes generated from the manufacturing process and converting them into raw materials for other products or finished products. The CE aims to improve the process of production and consumption by offering more

advanced packaging technologies, more straightforward packaging, more efficient home appliances, and a simpler lifestyle toward environmental protection goals.

According to Saavedra et al. [12], the CE model reduces waste, and the raw materials must be returned to the production process. Accordingly, CE policy researchers must focus on waste treatment, including process-based approaches to waste removal. Research by Hauschild et al. [15] also shows that opportunities to reduce waste in the production process and the product life cycle require manufacturers to reallocate resources, retrain workers, and reconfigure the machine system.

Dijk et al. [16] proposed a mixed policy to stimulate resource efficiency, emphasizing primary and complementary policy instruments such as raw material taxes, expanded producer liability, and other requirements of technical demand. Wilts and O'Brien [17] proposed a similar policy mix-based analytical approach to understanding better resources in the EU, such as the design of tools, policy synergies, and policy consistency. Hughes and Ekins [18] have also argued that mixed policies for efficient use of resources should be holistic and mutually reinforcing across policy domains and focus on win-win scenarios. Watkins et al. [19] provided qualitative environmental assessments of land use and other policies. However, this study also argued that cultural and behavioral change conditions are essential targets in monitoring and regulation.

The integrated policy-based analytical approach argues that no single policy can foster the complementarity between the sectors and sectors needed for the transition to a CE. Domenech and Bahn-Walkowiak [20] provide an overview of EU policies on the effective use of human resources for economic incentives and procedures in the context of CE adoption. The study finds that policy-binding goals must still focus primarily on the output side of resource flows (i.e., waste) while the input side is completely ignored or addressed through goals. Targets are ambitious, optional, and scattered throughout policy documents. The authors suggest that complex targeting will also affect the acceleration of the transition to a CE. Other studies on the CE transition through resource efficiency include raw materials in the Asia Pacific region, producer responsibility to delegate power, and responsibility for the industry. Wang and Zhou [21] also analyzed barriers to resource efficiency-based investment and stakeholder frameworks for implementing reverse logistics.

Gray [22] argued that the CE offers significant benefits in improving resource use efficiency, especially urban and industrial waste, and balancing economic, environmental, and social factors. A CE can optimize natural resources by improving efficiency and transitioning from open energy and material cycles to closed material cycles, as well as minimizing waste in industrial production. Furthermore, the inner circle economy is becoming an economic strategy rather than a purely environmental strategy. According to Feng and Yan [23], the CE is the new business model expected to bring more sustainable development to economies. Cainelli et al. [24] also show that the CE creates new business opportunities from recycled products and services. Reducing, reusing, recycling materials, as well as improving and innovating value chains and supply chains, will attract investors toward a cleaner industrial production method [23,25].

Over the years, studies on CE have been carried out, and many conceptual frameworks of CE have been formed. The 3R Principle is one of the schools within the framework of CE practice. The 3R has also been supplemented with other principles such as "recovery," translating into the broader concept of 4R as reduce, reuse, recycle and recover [26]. Jawahir and Bradley [27] proposed redesigning and redefining data to have a 6R framework that includes reducing, reusing, recycling, recovering, redesigning, and reproducibility to provide a closed product life cycle system as the basis for sustainable production. However, it has been suggested that further expansion of the "3R" frameworks may create some confusion in the documentation and application of heterogeneous concepts around the CE principles.

CE studies have also been carried out in many countries around the world. Levitzke's research [28] provides examples of CE development in South Australia. The CE builds on the 'reduce, reuse and recycle' hierarchy of wastes mainly in Australia's states and territo-

ries over the past decade [29]. Furthermore, South Australia's achievements in solid waste recycling and recovery are also central to developing a low-carbon economy. Employment opportunities related to developing aspects of the CE are highlighted [28]. The study also estimated the environmental and social impact of the CE by assessing 2030 greenhouse gas emissions and employment outcomes in Australia. The report describes the interdependence between 78 industries and shows that the output of one industry can also become the input of another [29]. The CE consists of material efficiency, renewables, and energy efficiency. Assumptions regarding "material efficiency" and "renewable and efficient" are made to quantify greenhouse gas emissions and employment impacts of the transition to a CE [30]. These assumptions relate to the life of the materials, the efficiency of renewable energy, and alternative fuels to fossil fuels.

In 2018, the Kenya Manufacturers Association introduced a plastic bottle recycling initiative to celebrate World Environment Day [31]. KAM's plastic bottle initiative has led to the establishment of a polyethylene terephthalate (PET) recycling company called PETCO Kenya which will ensure the sustainable management of plastic materials through recycling and reuse of water. All private domestic producers will be represented in PETCO to ensure the initiative's nationwide dissemination. PETCO Kenya aims to recover and recycle materials at a rate of 70% of total plastic materials by 2030. The recycling and reuse of plastic bottles in Kenya have led to business initiatives. Urban residents, especially those living in informal settlements, often use reusable plastic bottles to create miniature gardens with plants such as ginger, peppers, and onions.

Awareness of the CE is also one of the necessary factors for the transition to a CE. Research by Langen and Passaro [32] has assessed CEs and the level of awareness of three related groups: researchers, economists, and managers. This study developed comprehensive literature on CE perception to design questionnaires, compare results, and create a more detailed analytical and interpretive framework regarding the awareness of the transition to a CE. The study results show that all three stakeholders see CE as a "zero waste" economy and, in a broader sense, a model for redesigning the current socioeconomic status that utilizes multiple renewable materials. Managers focus more on realizing CE toward economic growth and job creation. The researchers expect increased environmental benefits from the transition to a CE. However, all three stakeholder groups share a vision that CE is in the early stages of transition. The researchers also suggest that a successful transition to CE depends on the management of the process. Researchers emphasize a top-down holistic approach to CE, while economists and managers follow a bottom-up approach [2]. This study also shows that implementing CE will be costly for governments, businesses, and users. Therefore, during this period, policy interventions played a leading role in disseminating positive perceptions about the concept and model of the CE.

Research by Almulhim and Abubakar [4] has shown that CE has been recognized globally as a sustainable development strategy of each country to face resource shortage or environmental pollution challenges. Although community behavior and lifestyles play an essential role in the transition to a CE, there is little research exploring the role and perceptions, attitudes, and community lifestyle during the transition to CE. Therefore, this study collected data on residents in the urban area of Saudi Arabia. Research results show that most people have limited awareness of CE. In addition, the study also indicates that CE awareness is positively related to educational attainment.

Similarly, in Vietnam, awareness of CE is still in its infancy, and empirical studies on the perception of CE are almost nowhere to be found. This gap research can be good for the authors when researching the factors affecting the transition to a CE in Vietnam.

## 3. Research Methodology

### 3.1. Research Design

This study uses the survey research method of a questionnaire and the structural equation model (SEM) as the analytical model. Structural linear model SEM is used to analyze multidimensional relationships between many variables in the model. The SEM is

suitable for long-term survey data sets and is used to estimate measurement and structural models of multivariable problems. SEM will show the relationship between latent variables. This relationship can make theoretical predictions that are useful to researchers. SEM combines all techniques, such as multivariable regression, factor analysis, and correlation analysis (between elements in the network diagram), to allow us to check the complex relationships in the model.

To implement the SEM model, we conducted data cleaning, scale testing and exploratory factor analysis (EFA). Finally, we used the SEM model to test the complex relationships between the variables in the model.

The questionnaire is divided into two parts. The first part contains basic information about the respondents such as age, gender, income and education. The second part is the Likert scale to measure the respondents' level of agreement to participate in the circular economy. The variables are scored from 1 to 5 according to the response level, respectively:

Strongly disagree: 1
Disagree: 2
No comment: 3
Agree: 4
Strongly agree: 5

Respondents are mainly from Hanoi and the surrounding areas of Hanoi. The authors chose Hanoi and its surrounding provinces due to its strong economic development. In addition, Hanoi and neighboring provinces are also home to many businesses, including manufacturing enterprises as well as service providers. The questionnaire used Google Forms and was collected through Facebook, email, social networks and even direct collection. The data collection period was approximately 60 days, including the time to distribute the test slips. The sample size meets the requirements of the SEM model.

The basic variables in the Likert scale are arranged as follows (Table 1):

**Table 1.** Interpretation of variables in the model.

| Encode | Explain |
|---|---|
| Attitude towards the decision (ATI) | ATI 1: I am very happy to be able to buy CE products/services<br>ATI 2: I am very proud to be able to buy products/services of the CE<br>ATI 3: I feel satisfied when I can buy CE products/services |
| Subjective norm (SJN) | SJN 1: I will buy these products if my family members and relatives will also buy these products<br>SJN 2: Opinions of experts and celebrities may influence my choice to participate in the CE<br>SJN 3: I will buy these products if my friends buy these products |
| Cognitive-behavioral control (CBC) | CBC 1: I have enough money to use the products of the CE<br>CBC 2: I am knowledgeable enough to participate in the CE<br>CBC 3: I can overcome barriers and prioritize participation in CE products |
| Benefit of individual economics (BOE) | BOE 1: I will stick with CE products/services if they regularly drop in price<br>BOE 2: I will buy CE products/services if they are cheaper than traditional products<br>BOE 3: I will support circular economies if they can help reduce costs and product costs |
| Attitude toward the environment (ATE) | ATE 1: I am willing to give up the products/services that I love to do if they harm the natural environment<br>ATE 2: I am willing to do good deeds for the environment without anyone knowing or thanking<br>ATE 3: I'm willing to do things that are good for the environment even if it's inconvenient<br>ATE 4: I often buy eco-friendly products<br>ATE 5: I will buy products/services that help reduce $CO_2$ emissions<br>ATE6: I usually buy green products |
| Integrated readiness to participate (IP) | IP 1: I will join the CE<br>IP 2: I will support activities to develop a CE in Vietnam<br>IP 3: I will recommend others to join the CE |

*3.2. Research Model*

The theory of planned behavior (TPB) is a model to explain human behavior under specific conditions. The intentions of different behaviors can accurately predict attitudes towards the behavior [33]. TPB is applied in this study to describe participants' behavior in the CE, such as environmentally friendly behavior and buying organic products. TPB will also be used to explain the intention of participants in the CE. The basic variables structured in this study include attitudes towards the decision to participate (IP) in the CE, subjective norm, perceived behavioral control, financial problems, and concerns about the environment [34].

### 3.2.1. Intention to Buy Environmentally Friendly Products (ATE)

This variable represents an individual's intention to buy green products or products that do not harm the environment. Green products include energy-saving and environmentally friendly products, use renewable energy, and do not use harmful chemicals. Many studies show that buying ecologically friendly products is one of the manifestations of a preference for participating in the CE. Liu et al. [35] analyzed that green purchasing behavior is one of the pieces of evidence showing people's participation in the CE. Buying green products and services is also considered a CE awareness. Thus, the need to buy green products shows people's concern for the environment, and therefore the proposed hypothesis is:

**Hypothesis 1 (H1):** *ATE has a positive covariate relationship with IP.*

### 3.2.2. Subjective Norms (SJN)

Subjective norms represent the degree of influence of those around them in making decisions or performing specific behaviors [36]. According to behavioral planning theory, social influence from family members, friends, and celebrities can discourage or encourage individuals to purchase and use new technology [37]. Therefore, the attitudes of family members and people will affect an individual's intention to buy green products. Research by Wu, Chen, Geng, Zhou, & Zhou [38] also shows that subjective norms promote the intent to protect the environment and strongly influence the intention to buy green products/services. If the people around an individual are willing to buy the products of the CE, the willingness to buy the product/service of that person also increases. Thus, hypothesis 2 is put forward as follows:

**Hypothesis 2 (H2):** *SJN has a positive relationship with IP.*

### 3.2.3. Financial Problems

This variable is related to the customer's finances, i.e., how much the customer is willing to pay for the product and service of new technology. Many studies on consumer behavior also show that cost significantly impacts consumer buying behavior. Choi and Parsa [39] show that price plays a decisive role in consumers' purchasing behaviors related to green products. Many studies show that consumers are price sensitive and are often not willing to pay extra for green products or healthcare services [40]. The main obstacle for many consumers is that they think green products usually cost more than conventional products. Reality shows that some green product features can provide other benefits to offset the higher purchase price. For example, electric cars are often more expensive than internal combustion engines, but maintenance and fuel costs are cheaper. Therefore, consumer financial issues can be one of the factors driving consumers' willingness to participate in the CE. Therefore, this hypothesis is expected as follows:

**Hypothesis 3 (H3):** *BOE has a positive relationship with IP.*

Attitude towards decision refers to the positive effect of specific emotions on certain behaviors. D-Jenne et al. [41] showed that emotional expectations, among other factors,

influence an individual's decision. Studies have also confirmed that individuals make choices based not only on utility levels but also on subjective expectations and experience a joyful feeling. In this study, emotions reflect the expectations of individuals when participating in the CE. Emotions influence people's choices [42], so people expect that participating in the CE will benefit the environment. Positive expectations will increase willingness to participate in the CE. This hypothesis is expected as follows:

**Hypothesis 4 (H4):** *ATI has a positive relationship with IP.*

3.2.4. Cognitive-Behavioral Control (CBC)

This variable subjectively evaluates an individual's acceptance level in performing a particular behavior. Factors such as resources or opportunities influence an individual's behavior. Research by Wiradhany et al. [43] showed that perceived behavioral control significantly affects the willingness to buy reusable products. When people have a positive view of behavior, the degree of willingness to accept the behavior is influenced by perceived behavioral control. Therefore, the author expects that perceived behavioral control will positively affect participation in the CE in this study (Table 2).

**Hypothesis 5 (H5):** *CBC has a positive relationship with IP.*

**Table 2.** Explanation of the participation of variables in the model.

| Variable | Variable Name Explain | Expectation |
|:---:|:---:|:---:|
| | Independent variables | |
| SJN | Subjective norm | (+) |
| ATI | Attitude towards decision | (+) |
| CBC | Cognitive-behavioral control | (+) |
| BOE | The benefit of individual economics | (+) |
| ATE | Attitude toward the environment | (+) |
| | Dependent variable | |
| IP | Integrated readiness to participate | |

## 4. Results

*4.1. Statistical Analysis*

The basic information is presented in Table 3. Among 431 respondents, the number of female respondents was superior, accounting for 73.3%, while male respondents accounted for 26.5%. Regarding the education level of the respondents, 67.7% were university graduates, 2.9% of the respondents graduated from vocational schools and only 4% graduated from high school. The number of respondents with a higher degree such as a master's or doctorate was also relatively high, accounting for about 25.6% of the sample. Regarding the average income, it can be seen that the common income is in the range of 5–10 million, which is consistent with the average income of the Vietnamese people.

Table 4 presents the mean and standard deviation of respondents in their willingness to participate in the circular economy. The results show that the average level of willingness to participate in the circular economy is 4.26.

**Table 3.** Basic statistics of the research sample.

| Variable | Categories | Percentage (%) |
|---|---|---|
| Gender | Male | 26.5 |
| | Female | 73.5 |
| Academic level | Diploma | 4 |
| | Vocational school | 2.9 |
| | Bachelor | 67.7 |
| | Postgraduate training | 25.6 |
| The average income | Under 3 million VND | 14.7 |
| | From 3 to 5 million VND | 10.7 |
| | From 5 to 10 million VND | 44.1 |
| | From 10 to 18 million VND | 12.3 |
| | From 18 to 32 million VND | 9.2 |
| | From 32 to 52 million VND | 4.5 |
| | Over 52 to 80 million VND | 2.5 |
| | Over 80 million VND | 2 |

**Table 4.** Descriptive statistics.

| | N | Minimum | Maximum | Mean | | Std. Deviation | Skewness | | Kurtosis | |
|---|---|---|---|---|---|---|---|---|---|---|
| | Statistic | Statistic | Statistic | Statistic | Std. Error | Statistic | Statistic | Std. Error | Statistic | Std. Error |
| SEX | 431 | 1 | 2 | 1.74 | 0.021 | 0.442 | −1.072 | 0.118 | −0.856 | 0.235 |
| AGE | 431 | 1 | 8 | 3.21 | 0.085 | 1.775 | 0.831 | 0.118 | −0.610 | 0.235 |
| IP1 | 431 | 1 | 5 | 4.13 | 0.040 | 0.831 | −0.516 | 0.118 | −0.566 | 0.235 |
| IP2 | 431 | 1 | 5 | 4.41 | 0.036 | 0.741 | −1.094 | 0.118 | 0.820 | 0.235 |
| IP3 | 431 | 1 | 5 | 4.25 | 0.039 | 0.803 | −0.787 | 0.118 | −0.023 | 0.235 |
| Valid N (listwise) | 431 | | | | | | | | | |

### 4.2. SEM Results

To be able to perform the verification steps for SEM, it is very necessary to assess the reliability of the scale.

The Cronbach's alpha of the scales in Table 5 shows that the reliability coefficients are satisfactory. Cronbach's alpha coefficient is used to eliminate the garbage variable first. Variables with an item-total correlation less than 0.3 will be excluded, and the scale must have an alpha reliability of 0.60 or higher [44]. After that, the variables with factor loading less than 0.50 in EFA will continue to be excluded. Therefore, all scales meet the reliability requirements (0.6 < 0.95) and are accepted, included in exploratory factor analysis (EFA) to test convergent and discriminant validity. The results of the scale reliability analysis show that the reliability coefficients of Cronbach's alpha of all scales are greater than 0.8.

After checking the reliability of the scale, exploratory factor analysis was conducted. The extraction method chosen for factor analysis is the principal components method with Promax rotation.

KMO coefficient and sig coefficient: The exploratory factor analysis for the independent variables shows that the *p*-value = 0.000 of Bartlett's test allows us to safely reject the null hypothesis H0 (H0: factor analysis does not fit the data). The KMO index = 0.926 shows that the model's relevance is high (Table 6).

The rotation factor matrix table shows that there are 6 factors (Table 7) extracted from the data to ensure adequate conditions to implement SEM.

**Table 5.** Testing the reliability of the scale.

| Variable | Scale Mean If Item Deleted | Scale Variance If Item Deleted | Corrected Item—Total Correlation | Squared Multiple Correlation | Cronbach's Alpha If Item Deleted |
|---|---|---|---|---|---|
| | | Attitude towards decision (Cronbach's Alpha = 0.888) | | | |
| ATI 1 | 8.36 | 2.430 | 0.782 | 0.620 | 0.841 |
| ATI 2 | 8.36 | 2.280 | 0.802 | 0.625 | 0.824 |
| ATI 3 | 8.44 | 2.456 | 0.762 | 0.584 | 0.858 |
| | | Subjective norm (Cronbach's Alpha = 0.823) | | | |
| SJN 1 | 7.60 | 3.559 | 0.655 | 0.455 | 0.780 |
| SJN 2 | 7.66 | 3.632 | 0.635 | 0.426 | 0.798 |
| SJN 3 | 7.83 | 3.176 | 0.749 | 0.562 | 0.682 |
| | | Cognitive behavioral control (Cronbach's Alpha = 0.838) | | | |
| CBC 1 | 7.52 | 2.677 | 0.690 | 0.479 | 0.786 |
| CBC 2 | 7.41 | 2.574 | 0.722 | 0.525 | 0.754 |
| CBC 3 | 7.39 | 3.039 | 0.699 | 0.497 | 0.782 |
| | | Benefit of indivudual economic (Cronbach's Alpha = 0.845) | | | |
| BOE 1 | 8.41 | 2.833 | 0.692 | 0.493 | 0.804 |
| BOE 2 | 8.35 | 2.417 | 0.763 | 0.589 | 0.737 |
| BOE 3 | 8.13 | 3.038 | 0.694 | 0.502 | 0.805 |
| | | Attitute for the environment (Cronbach's Alpha = 0.863) | | | |
| ATE 1 | 20.92 | 10.439 | 0.601 | 0.377 | 0.852 |
| ATE 2 | 20.59 | 11.480 | 0.560 | 0.374 | 0.856 |
| ATE 3 | 20.92 | 10.749 | 0.685 | 0.485 | 0.835 |
| ATE 4 | 21.00 | 10.131 | 0.735 | 0.603 | 0.825 |
| ATE 5 | 20.81 | 10.619 | 0.731 | 0.575 | 0.828 |
| ATE 6 | 21.01 | 10.540 | 0.645 | 0.496 | 0.842 |
| | | Intergrated Ready to Participate (Cronbach's Alpha = 0.814) | | | |
| IP 1 | 8.66 | 1.930 | 0.657 | 0.431 | 0.756 |
| IP 2 | 8.39 | 2.143 | 0.672 | 0.451 | 0.742 |
| IP 3 | 8.54 | 1.979 | 0.673 | 0.454 | 0.738 |

**Table 6.** KMO coefficient and Bartlett's test for factors.

| Kaiser–Meyer–Olkin Measure of Sampling Adequacy | | 0.926 |
|---|---|---|
| Bartlett's Test of Sphericity | Approx. Chi-Square | 5094.924 |
| | df | 210 |
| | Sig. | 0.000 |

**Table 7.** Pattern matrix.

| | Factor | | | | | |
|---|---|---|---|---|---|---|
| | 1 | 2 | 3 | 4 | 5 | 6 |
| ATE 5 | 0.920 | | | | | |
| ATE 4 | 0.807 | | | | | |
| ATE 6 | 0.666 | | | | | |
| ATE 3 | 0.537 | | | | | |
| ATI 1 | | 0.907 | | | | |
| ATI 2 | | 0.843 | | | | |
| ATI 3 | | 0.776 | | | | |
| BOE 2 | | | 0.988 | | | |
| BOE 3 | | | 0.695 | | | |
| BOE 1 | | | 0.620 | | | |
| SJN 3 | | | | 0.949 | | |
| SJN 1 | | | | 0.698 | | |
| SJN 2 | | | | 0.644 | | |
| CBC 2 | | | | | 0.858 | |
| CBC 3 | | | | | 0.684 | |
| CBC 1 | | | | | 0.677 | |
| IP 2 | | | | | | 0.741 |
| IP 3 | | | | | | 0.735 |

### 4.3. SEM Test Results

We use AMOS to assess the SEM. The model's fit index includes basic indexes such as chi-square ($\chi^2$), degrees of freedom (df), $\chi^2$/df ratio, root-mean-square error of approximation (RMSEA), comparative fit index (CFI), goodness of fit (GFI), and adjust goodness of fit (AGFI) [45].

In hypothesis testing and the research model, the SEM linear structural model has many advantages over multiple regression analysis methods and multivariable regression because it can calculate measurement error. Therefore, the SEM analysis method has been widely used in social sciences in recent years.

The CFA results show chi-squared/df = 2.730 with $p$ = 0.000, TLI = 0.940, CFI = 0.953 and RMSEA = 0.063 < 0.08. These indicators show that the model fits the data (Figure 1). The correlation coefficients between the concepts show that these coefficients are less than 1 (statistically significant).

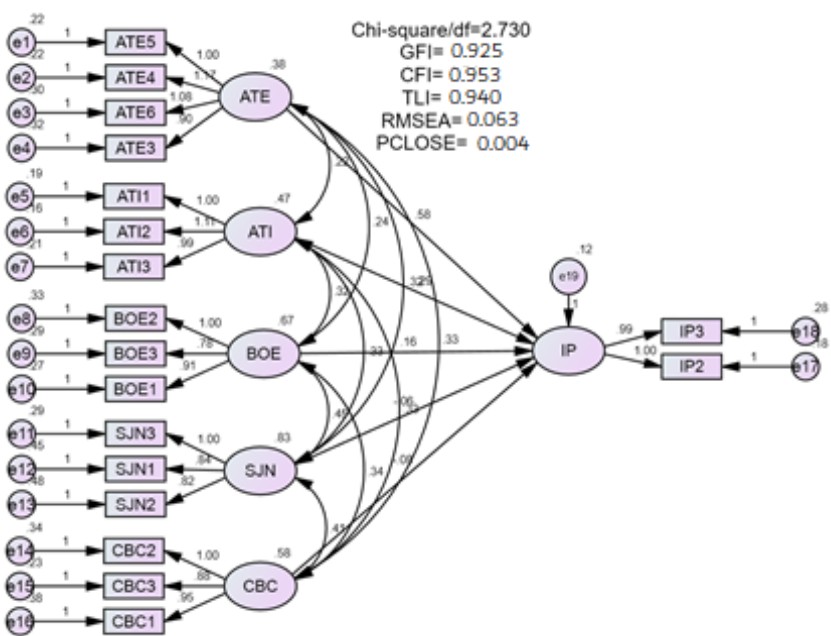

**Figure 1.** SEM model results with 6 factors.

Convergence, reliability, and discriminant testing are essential in CFA. If the factors do not ensure convergence and reliability, the data's meaning cannot be expressed. According to Hair et al. [45], a value of CR (composite reliability) > 0.7 is satisfactory. Thus, in the research sample, all factors have CR > 0.7 and reach the confidence level (Table 8). However, there are three variables, ATE, ATI, and BOE, that meet the criteria for convergence. The study sample also met the requirements of discriminant with maximum shared variance (MSV) < average variance extracted (AVE).

**Table 8.** Results of testing the fit of the research model.

|     | CR | AVE | MSV | ASV | SJN | ATE | ATI | BOE | CBC |
|-----|-----|-----|-----|-----|-----|-----|-----|-----|-----|
| **SJN** | 0.705 | 0.464 | 0.371 | 0.333 | 0.681 | | | | |
| **ATE** | 0.859 | 0.553 | 0.480 | 0.326 | 0.574 | 0.744 | | | |
| **ATI** | 0.829 | 0.572 | 0.402 | 0.317 | 0.524 | 0.522 | 0.756 | | |
| **BOE** | 0.773 | 0.501 | 0.371 | 0.303 | 0.609 | 0.470 | 0.566 | 0.703 | |
| **CBC** | 0.710 | 0.471 | 0.480 | 0.384 | 0.596 | 0.693 | 0.634 | 0.547 | 0.686 |

The ATE hypothesis positively affects the willingness to participate in the CE. From the normalized beta weights, we see that the impact of the ATE variable on IP is the largest with statistical significance above 95% ($\beta$ = 0.578; sig = 0.000 < 0.05). From this, we conclude

that environmental concerns have the greatest impact on willingness to participate in the CE. Therefore, the hypothesis is accepted (Table 9).

**Table 9.** Results of impact assessment of relationships.

|  | Estimate | S.E. | t-Values | *p*-Values | Results |
|---|---|---|---|---|---|
| IP ← ATE | 0.578 | 0.075 | 7.729 | 0.00 | Accepted |
| IP ← ATI | 0.288 | 0.058 | 4.953 | 0.00 | Accepted |
| IP ← BOE | 0.163 | 0.050 | 3.290 | 0.001 | Accepted |
| IP ← SJN | −0.058 | 0.046 | −1.247 | 0.212 | Not accepted |
| IP ← CBC | −0.076 | 0.066 | −1.150 | 0.250 | Not accepted |

The hypothesis is that ATI (attitude towards decision) positively impacts the willingness to participate in the CE. From the normalized beta weights, we see that ATI positively affects IP with a statistical significance above 95%. (β = 0.288; sig = 0.000 < 0.05). From there, we conclude that the attitude toward the decision positively impacts willingness to participate in the CE. Therefore, the hypothesis is accepted (Table 9).

The financial problems hypothesis (BOE) positively affects the willingness to participate in the CE. From the normalized beta weights, we see that ATI positively affects IP with a statistical significance above 95% (β = 0.163; sig = 0.001 < 0.05). From that, we conclude that financial factors positively impact willingness to participate in the CE. Therefore, the hypothesis is accepted (Table 9).

The subjective norm hypothesis (SJN) positively affects willingness to participate in the CE. However, we see that SJN has a negative effect on IP with a statistical significance above 95% (β = −0.058; sig = 0.212 > 0.05). From that, we conclude that the subjective norm factors have a negative impact on the willingness to participate in the CE. Therefore, the hypothesis is not accepted (Table 9).

The cognitive-behavioral control hypothesis (CBC) positively affects the willingness to participate in the CE. However, we see that CBC has a negative effect on IP with statistical significance above 95% (β = −0.076; sig = 0.250 > 0.05). From there, we conclude that perceived behavioral control factors have a negative impact on the willingness to participate in the CE. Therefore, the hypothesis is not accepted (Table 9).

## 5. Discussion and Conclusion

This study examines the factors affecting people's willingness to participate in the CE in Vietnam. Experimental results show that ATI has a significant effect on IP. In today's social context, individuals are always looking for personal identity in their groups. They tend to be influenced by each other to match the rest of the group they are in. Thus, the opinions of individuals influence other individuals. Mkhize and Ellis [46] tried to adopt environmentally friendly behaviors such as buying green products. Therefore, ATI has a significant impact on purchasing CE products. The results of this study are also consistent with previous research showing that the attitude towards a decision significantly affects the level of willingness to buy CE products, which also meant that they were willing to participate in the CE. Therefore, relevant policymakers should carefully consider promoting individuals' participation in the CE. Policymakers need to pay more attention to people's groups such as family, friends, and mainstream media. Another policy implication is that it is necessary to distinguish population groups according to different criteria such as occupation, education level, and age to assess people's willingness to participate in the CE.

The research results also show a positive relationship between the level of willingness to participate in the CE and concern for the environment (ATE). Individuals who have environmental sacrifice are more likely to express their intention to buy green products. Kurrbanov et al. [47] stated that concern for the environment is a typical attitude towards the environment. A positive attitude towards the environment also encourages individuals to look for options for eco-friendly products and fit daily necessities. Cardoos et al. [48,49] also showed individuals' level of willingness to sacrifice money, level of willingness to

sacrifice life, and level of willingness to pay taxes to achieve the goal of environmental protection. As the willingness to sacrifice money to protect the environment increased, the level of payment for green products also increased. Individuals in this category will also be more willing to replace products of the linear economy with CE products. They will also be glad to bear the effects of this change. From these two aspects, the increase in environmental protection will lead to the rise of ATE. Environmental protection attitudes also play a guiding role in purchasing and selling green products. Based on these findings, policymakers need to develop appropriate policies to promote the spirit of dedication of the people. The government needs to balance practical difficulties with efficiency to develop appropriate ways to encourage people to participate more deeply in the CE. Key stakeholders involved in the CE include consumers, governments, and businesses. CE consumption is also an essential aspect of assessing people's participation in the CE. One of the purposes of the CE is to limit the use of non-renewable materials and increase the use of renewable materials, converting waste into raw materials. Muranko et al. [25] argued that consumers buying and selling green products is indispensable to the CE. The intention or need to buy and sell products in a CE drives businesses to produce products and services in a CE. Individuals' purchasing behavior of CE products can be called participating in the CE. Therefore, the greater the intention to buy green products, the higher the willingness to participate in the CE.

The BOE that impacts IP also shows that individuals intend to purchase CE products when they perceive these products/services to provide more excellent economic benefits. Individuals always seek to maximize their utility when making consumption-related decisions. Research by Lee and Chen-Yu [50] has shown that the lower the product price, the higher the purchase intention. If consumers perceive the price of green products to be lower than that of traditional products with the same usability, they will choose CE products. Thus, previous studies have shown that price, discount, and other economic factors positively impact consumer purchase intention. Therefore, the task posed to policymakers is the pricing policy for products of the CE. Policymakers need to have policies to support prices, adjust tax rates, and support production supply chains to reach the ultimate goal of reducing production costs and thus stimulating consumer interest in CE products.

However, the limitation of the present study is that the factors affecting participation in the circular economy are still few. There are still many other factors that can affect the willingness of individuals and businesses to participate in the circular economy.

**Author Contributions:** Conceptualization, T.V.T. and A.T.T.L.; methodology, T.M.T.; software, T.M.T.; validation, T.V.T. and T.H.P.; formal analysis, T.M.T.; investigation, T.V.T. and T.H.P.; resources, T.M.T. All authors have read and agreed to the published version of the manuscript.

**Funding:** This research received no external funding.

**Informed Consent Statement:** Informed consent was obtained from all subjects involved in the study.

**Data Availability Statement:** The data presented in this study are available on request from the corresponding author.

**Conflicts of Interest:** The authors declare no conflict of interest.

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
