# Peer review of "Evaluation of Factors Affecting the Transition to a Circular Economy (CE) in Vietnam by Structural Equation Modeling (SEM)"

_sustainability, doi:10.3390/su14020613_

Round 1
Reviewer 1 Report
Review Report: Evaluation of factors affecting the transition to a circular econ-omy (CE) in Vietnam by Structural Equation Modeling (SEM)
Summary
This paper examines factors affecting the transition to a circular economy (CE) in Vietnam and is based on the survey and the Structural Equation Modeling. The paper is well written, however needs some improvements to be published.
- Main Comments and Suggestions
Is the content succinctly described and contextualized with respect to previous and present theoretical background and empirical research (if applicable) on the topic?
Yes, it is. The Authors give the theoretical framework and present the results of the literature review concerning the idea of a circular economy as well as the theory of planned behavior (TPB).
Are the research design, questions, hypotheses, and methods clearly stated?
The paper clearly presents hypotheses development and research aims. However, methods used by the Authors are not clearly described. They present their survey questionnaire in detail however there is no information about the Structural Equation Modeling and the statistical tests used.
Are the arguments and discussion of findings coherent, balanced, and compelling?
The arguments are convincing, and the discussion of findings is related to the research results of other authors. In my opinion, the authors should indicate the limitations of their research and propose directions for further research.
For empirical research, are the results clearly presented?
No, the results are not clearly presented. The Authors divided the presentation of the results into sections. However, the presentation of the results was made mainly in the form of a tabular presentation of the data, but there are no more precise explanations as to how the obtained results should be understood.
Secondly, in the 3.1. Questionnaire: “The authors distributed 450 questionnaires and collected 431 pivot questionnaires to conduct the study.” This is a very brief description of the research sample. In my opinion, it is incomplete. I would like to know how the questionnaire was distributed, where, according to what selection criterion, etc. It is only in point 4.1. Statistical analysis that the Authors indicate that the respondents were Hanoi residents. In my opinion, the description of the research sample must be in point 3. Research methodology.
Finally, point 4.1. Statistical analysis does not provide any statistical analysis but gives information about personal data of the respondents.
Is the article adequately referenced?
Yes, it is. The literature review is well written. There is a clear literature review in the separate section.
Are the conclusions thoroughly supported by the results presented in the article or referenced in secondary literature?
The conclusions are mainly limited to a description of results. However, there are some references to the results in the literature. The Authors did not indicate any limitations or possibilities for further research directions.
Hope the above comments will help the Authors to further develop the paper.
Author Response
Dear Reviewer,
First of all, we would like to thank you for your valuable comments. Thanks to the above comments, we could correct and improve our journal articles.
All of the correct and improvements you can find here in the attachment.
Finally, thank you for your comments, and I wish you a wonderful New Year.
Thanks and best regards,

Reviewer 2 Report
Dear Authors
Initially, greeting and wishing to meet you in good health. I wish you and all your families an excellent New Year's Eve. As for the analyzed paper, I can assure you that the research object is current and of great relevance to academia and society. You have structured the article precisely. They built a good theoretical framework, which supports the arguments. The method was well used and suited to the type of study. The conclusion was well elaborated and points out the results of the research. References are current. I don't have any opportunity for improvement right now. In this way, I am of the opinion, unless better judged, that the paper meets minimum publication conditions. My opinion is for acceptance.
I wish you success in your careers.
Regards
Author Response
Dear Reviewer,
Firstly, we wish you and your families an New Year's Eve with many happiness and success.
We are happy with your comments. However, we have also corrected and upgraded the article to meet the requirements of the journal. Once again, many thanks for your report.
Best regards,